# Developing the evidence base for evaluating dementia training in NHS hospitals (DEMTRAIN): a mixed-methods study protocol

Faraz Ahmed [1], Hazel Morbey,[1] Andrew Harding,[1] David Reeves,[2]
Caroline Swarbrick,[1] Linda Davies [2], Mark Hann,[2] Fiona Holland,[2] Ruth Elvish,[3]
Iracema Leroi,[4] Simon Burrow,[3] Alistair Burns,[4] John Keady,[3] Siobhan Reilly [1]

¹Division of Health Research, Faculty of Health and Medicine, Lancaster University, Lancaster, UK
²Division of Population Health, Health Services Research & Primary Care, The University of Manchester, Manchester, UK
³Division of Nursing, Midwifery & Social Work, University of Manchester, Manchester, UK
⁴Division of Neuroscience & Experimental Psychology, University of Manchester, Manchester, UK

**Correspondence to**
Dr Siobhan Reilly;
s.reilly@lancaster.ac.uk

## ABSTRACT

**Introduction** Around 70% of acute hospital beds in the UK are occupied by older people, approximately 40% of whom have dementia. Improving the quality of care in hospitals is a key priority within national dementia strategies. Limited research has been conducted to evaluate dementia training packages for staff, and evaluation of training often focuses on immediate, on-the-day training feedback and effects.

**Objectives** Our study aims to answer two research questions: (1) How do variations in content, implementation and intensity of staff dementia training in acute hospitals in England relate to health service outcome/process measures and staff outcomes? and (2) What components of staff dementia training are most strongly related to improved patient and staff outcomes?

**Methods and analysis** Using the principles of programme theory, a mixed-method study will be used to identify mechanisms and the interactions between them, as well as facilitators and barriers to dementia training in hospitals. We will use existing data, such as Hospital Episode Statistics, alongside two surveys (at hospital and staff level).We will recruit up to 193 acute hospitals in England to participate in the hospital level survey. We aim to recruit up to 30 staff members per hospital, from a random sample of 24 hospitals. In addition, we will explore the cost-effectiveness of dementia training packages and carry out an in-depth case study of up to six hospitals.

**Ethics and dissemination** The study has been reviewed and approved by the Faculty of Health and Medicine Research Ethics Committee (FHMREC 17056) and Health Research Authority (Integrated Research Approval System (IRAS) ID 242166: REC reference 18/HRA/1198). We plan to develop both standard (eg, academic publications, presentations at conferences) and innovative (eg, citizen scientist web portals, online fora, links with hospitals and third sector organisations) means of ensuring the study findings are accessible and disseminated regionally, nationally and internationally.

## Strengths and limitations of this study

► This study will use a range of existing datasets (including Hospital Episode Statistics and National Audit of Dementia's Organisational Checklist) and mixed-methods approaches to make recommendations about dementia training in acute hospital settings.

► Using the principles of programme theory, we will expand on mechanisms and the interactions between them, as well as facilitators and barriers to dementia training in the hospitals.

► This study aims to explore the long-term impact of dementia training (if any) on health service outcomes/process measures (eg, length of stay, emergency readmissions and mortality) and staff outcomes (eg, confidence, staff strain, knowledge of dementia).

► Given the multiple initiatives and dementia policies employed in hospitals across England, it will be difficult to establish the effects of dementia training directly on patient and staff outcomes.

Acute National Health Service (NHS) Trusts provide a range of services, including accident and emergency departments, inpatient and outpatient medicine and surgery and in some cases very specialist medical care.[4] They provide secondary care, ranging from relatively small district hospitals to large city teaching hospitals in England. The term acute, generally refers to physical illnesses and conditions, which are usually short term and require diagnostic tests, treatment and follow-up care.[4 5] At any one time, around 70% of acute hospital beds are occupied by older people, 40% of whom have dementia.[6–8] Despite limited literature in the area, a recent ethnography of the care received by people living with dementia in acute hospital suggest that staff struggle to respond to the needs of people living with

## INTRODUCTION

Improving the quality of care in general hospitals continues to be one of the key priorities within national dementia strategies.[1–3]

dementia in acute care settings and that training may be one of the intervention that can help to address quality of care issues observed.[9] Dementia awareness and care training is widely thought to increase staff competencies,[10 11] although types of training and how effectiveness is assessed is highly variable.

Since 2013, Health Education England has overseen increases in the provision of foundational-level dementia training.[12 13] However, limited research has so far been conducted to evaluate dementia training packages. Evaluation tends to focus on immediate, on-the-day training feedback and effects. Moreover, training for staff in any care setting is seldom evaluated for its impact on the care of people living with dementia and for long-term effects and impact on staff.[14 15] Given the high variability of dementia training packages,[11] there is a need to better understand the current evidence and context of dementia training that is provided in acute NHS hospitals. This study will explore the complex workings of dementia training, including exploring relationships with patient and staff outcomes. This mixed-methods study aims to address two overarching research questions:

1. How do variations in content, implementation and intensity of dementia training in hospitals in England relate to health service outcome/process measures (eg, length of stay (LoS), emergency readmissions (ERs) and mortality) and staff outcomes (eg, confidence, staff strain, knowledge of dementia)?
2. What components of dementia training are most strongly related to improved patient and staff outcomes?

The findings will lead to a set of specific recommendations and guidance about which elements of training packages are more effective and appropriate for use in acute hospitals.

## METHODS AND ANALYSIS
The three phases of the study are:
► Phase 1: Mapping the evidence base for dementia training in hospitals.
► Phase 2: Cohort study to assess differences in acute NHS hospital outcomes for people living with dementia compared with those without, and the impact of training methods on patient outcome.
► Phase 3: Multiple hospital case studies of staff dementia training and care in acute NHS hospitals.

Our mixed-methods study will use the principles of programme theory[16] to identify mechanisms and the interactions between them, as well as the facilitators and barriers to dementia training in the hospital. Developing the evidence base for evaluating dementia training in NHS hospitals (DEMTRAIN), one of eight work programmes (WP) within the Neighbourhoods and Dementia Study (funded by the Economic and Social Research Council and the National Institute for Health Research), is developing the evidence base for evaluating dementia training in acute NHS hospitals. The DEMTRAIN study aims to integrate the various phases, which includes quantitative and qualitative data (see figure 1).

### Phase 1: mapping the evidence base for dementia training in hospitals
#### Aims
1. To review the current evidence base for staff dementia training in hospital settings.
2. To map the variation in staff dementia training currently provided to acute hospital staff in England.
3. To assess the impact of differences in staff dementia training on staff knowledge, practice, organisational culture and staff strain in acute NHS hospitals.

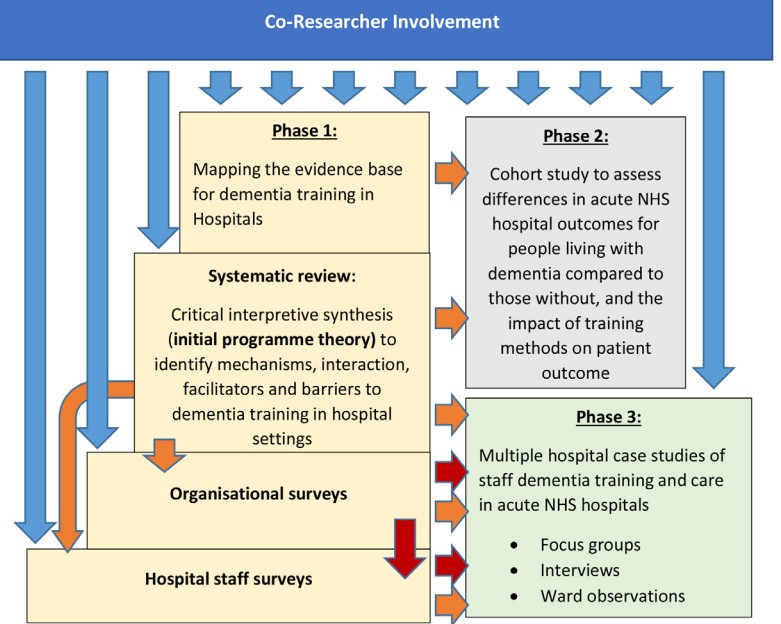
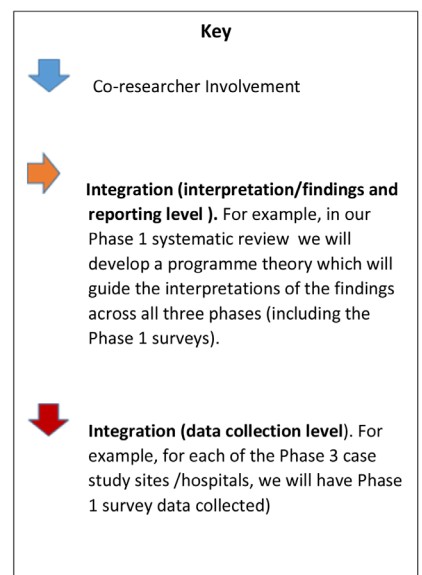

**Figure 1** DEMTRAIN integration of co-researcher involvement, study phases, data collection and findings. DEMTRAIN, Developing the evidence base for evaluating dementia training in NHS hospitals; NHS, National Health Service.

## Methods

Phase 1 will involve three main elements:

I. We will conduct a systematic review of the evidence base for training hospital staff in dementia awareness and care exploring both the quality and effectiveness of training programmes. This will be achieved by conducting a critical interpretive synthesis,[17] which will form the basis of a logic model using the principles of programme theory to identify mechanisms, interaction, facilitators and barriers to dementia training in hospital settings. The protocol for this element of phase 1 is registered on PROSPERO and is not outlined here.[18]

II. We will conduct an organisational survey in acute NHS hospitals in England (approximately 193 hospitals will be approached using the College Centre for Quality Improvement (CCQI) National Audit of Dementia (NAD) sample frame contact list) to identify training strategies; establish the presence, frequency, duration, characteristics and format of training programmes; quality of training and resources used in developing training. We will also explore what other initiatives are implemented (eg, having a dementia champion, active participation in Butterfly scheme, Dementia Friends, carers allowed on wards at meal times).

III. We will conduct a survey of staff (hospital staff survey) to collect data on the extent of training, knowledge, attitudes, staff strain, organisational culture and satisfaction in caring for patients living with dementia. We aim to recruit 24 hospitals randomly sampled from stratified lists of hospitals scoring low, mixed and high across three main domains of dementia care (governance, staff training and patient care) derived from the NAD in 2016).

### Recruitment for organisational survey in acute NHS hospitals in England

We will use NAD data to identify potential hospitals for the hospital organisational survey. Our preliminary discussions with our advisory group (including representatives from NAD who are involved in similar surveys, and local dementia leads who are hospital staff responsible for ensuring that their hospital is aligned with NHS Trust and National dementia strategies, and involved in developing and implementing dementia training in their hospitals), suggests that in most cases dementia leads at each hospital will have access to the data needed to complete the organisational level survey. However, given the segmented and large variations in recording systems for dementia training across Hospitals in England, the information needed to complete the survey may be held in other departments in the hospital, such as other Information Technology (IT) or human resource systems. On this basis, we will have multiple survey collection approaches (ie, online and telephone completion), with researcher-led support to ensure high-quality data and completion. Figure 2 presents the hospital recruitment process for the organisational survey.

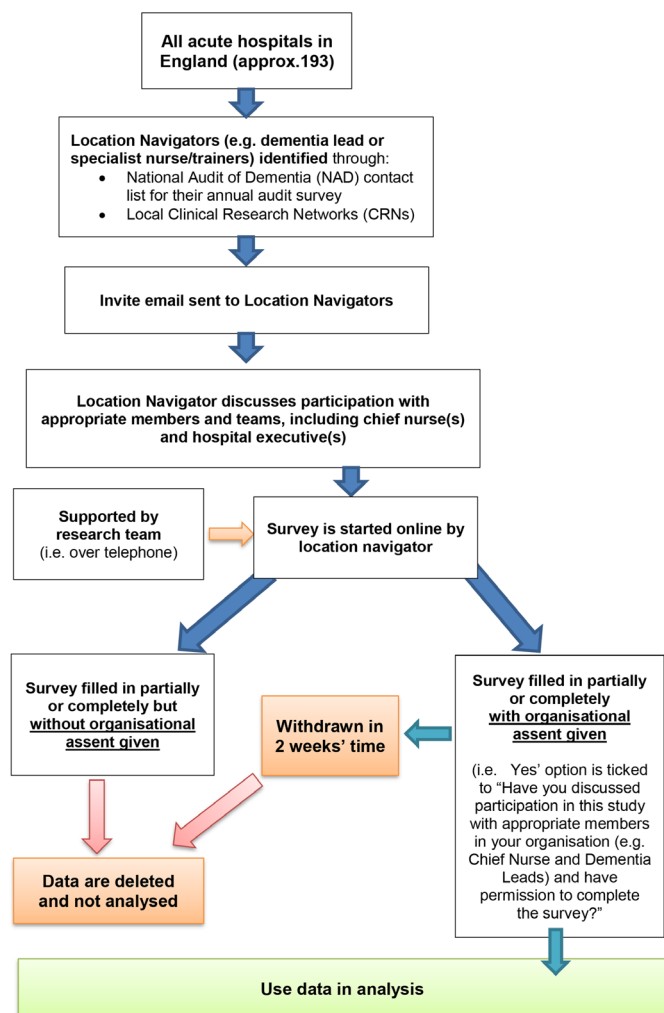

**Figure 2** Recruitment process for acute NHS hospitals taking part in the DEMTRAIN Hospital organisational survey of dementia training and care in England. DEMTRAIN, Developing the evidence base for evaluating dementia training in NHS hospitals; NHS, National Health Service.

Dementia leads will be responsible for completing the data. However, in cases where the information needed to complete the Hospital Organisational Survey is not directly available to the dementia leads, the dementia leads will act as location navigators and will direct requests to complete the survey to the appropriate person in the acute NHS hospital.

Before starting the organisational survey, participants will be asked if they have discussed participation in this study with appropriate members in their organisation, and that their hospital and or Trust has given assent to participate; where assent is not given, survey data will be deleted and not analysed.

### Recruitment for staff survey in two specific wards in acute NHS hospitals in England

Using the NAD returns for 2016, we will combine across sets of related question items to construct performance scores for each hospital on three main domains of care for inpatients with dementia: governance; staff training

and patient care (policies/processes concerning nutrition, discharge, assessment, communication and carers' rating of patient care). We will then identify subgroups of hospitals scoring high, low and mixed across the three domains. Within each of these subgroups hospitals, we will randomly sample for invitation into the survey, with the objective of recruiting eight hospitals from within each subgroup (24 in total).

We will survey different types of staff members working on two contrasting acute wards where dementia is expected to be prevalent:

▶ Ward 1: elderly care/older people's ward (if the acute NHS hospital does not have these wards, an orthopaedic ward will be selected instead).

▶ Ward 2: general surgery or surgical ward (if the acute NHS hospital does not have these wards, an oncology ward will be selected instead).

We will recruit staff from five categories in each acute NHS hospital:

1. Doctors.
2. Nurses.
3. Healthcare assistants.
4. Allied healthcare professionals (eg, physiotherapists, dieticians).
5. Support staff in the hospital (eg, housekeepers, porters, receptionists).

In view of our member involvement (alternatively termed as patient and public involvement) and advisory group feedback, our sample will focus on staff categories with more frequent interactions with patients living with dementia. It is thought that the above staff groups may be more likely to affect patient-related processes of care and outcomes. Staff will be sent electronic or postal copies of the staff survey depending on individual preference.

Location navigators and local clinical research networks (CRNs) will help identify 15–30 participants from each hospital. The research team will inform location navigators and CRN to recruit staff in a particular ward on a particular day and to select a random sample from ward/hospital staff lists/registers (eg, every fourth staff member on the list will be approached). However, this may not be possible in the absence of staff lists/registers. Data on representativeness of the sample in relation to the ward will also be collected (ie, estimated number of staff on ward on a given day, in comparison to the number of surveys completed).

### Phase 2: cohort study to assess differences in acute NHS hospital outcomes for people living with dementia compared with those without, and the impact of training methods on patient outcome
#### Aim
To determine the types and elements of training that are most strongly associated with hospital outcomes for people living with dementia, including length of stay, (ERs within 30 days of prior discharge) and mortality (death within 30 days of discharge) for older people.

#### Methods
The primary source of information on hospital outcomes will be the Hospital Episode Statistics (HES) database, which holds routinely collected information on all inpatient spells in England and Wales. Outcomes for patients with a diagnosis of dementia will be compared with outcomes for patients without a diagnosis, with a focus on how hospital training scores relate to the difference in outcomes between these two groups. Comparisons will be made separately for the financial years 2010/2011, 2012/2013 and 2016/2017 to coincide with data on staff training and other dementia initiatives in hospitals collected under the NAD (see below). Patients living with dementia will be defined as those with a diagnosis of dementia reported in the hospital record at any time over the previous 5 years; patients without a diagnosis will be used as controls. This approach has been used in several previous studies (eg, Care Quality Commission, State of health care and social care in England in 2012/2013 Technical Annex 3)[19] and the rate of misclassification is believed to be acceptably small.[20] Corresponding NAD data for 2010/2011, 2012/2013 and 2016/2017 will be used to construct hospital-level scores on a number of key domains of dementia care, in particular governance, staff training and patient care.

#### Details of additional secondary research data
We will use other relevant secondary data where appropriate. In addition, we will develop a number of collaborations and explore other sources of data, such as data already available in the public domain or routinely collected as part of service evaluation. These may include further information on care, details and rates of staff training and patient experience among people living with dementia.

### Phase 3: multiple hospital case studies of staff dementia training and care in acute NHS hospitals
#### Aims
1. To determine the content, implementation and intensity of training in contrasting hospitals and to explore the acceptability and feasibility of training to a range of hospital staff.
2. To identify barriers and facilitators for the implementation, and sustainability of staff dementia training in acute NHS hospitals.
3. To identify lessons about the implementation of staff dementia training that might be applicable in other care settings.

#### Methods
Case study approaches in health research tend to involve multiple methods and offer pragmatic and appropriate approaches to provide an in-depth and strong focus on being able to report on how findings are contingent on context.[21 22] Observational methods further enhance the possibility for insights in case study research.[23] Our design includes multiple data collection approaches, including observations, focus groups and interviews within each case study hospital. This phase of the study

is predominately qualitative with data collection from multiple sources with a range of stakeholders within each hospital. We will use the principles of programme theory and realist approach to identify theories of change with respect to training initiatives in different hospital contexts. For instance to identify mechanisms and the interactions between them to explain what has caused an outcome and in what context,[24 25] as well as facilitators and barriers to dementia training in the hospital and its related outcomes.[24] Programme theory, within the theory-driven evaluation field, refers to a variety of ways of describing and evaluating a programme or intervention (such as staff dementia training) by developing an understanding of the causal modal that connect the inputs and activities of an intervention to its outcomes.[16]

Using an organisational case study design, we will explore the implementation of training practices in hospitals.[26] Using data from phase 1, we will select up to six contrasting hospitals based on a number of factors related to the implementation of training such as: intensity, duration, coverage of training; geography (Sustainability and Transformation Plans region and hospital type) (see table 1). A series of semistructured interviews and focus groups will be conducted with a purposive sampling strategy to recruit staff with different levels of training and experience (including, doctors, nurses, support and domestic staff). These interviews and focus groups will explore expectations and experiences of staff dementia training among hospital staff. In addition, researchers will observe staff activity on wards at up to two shift changeovers. Observations will focus on administrative/general duties and the work environment related to dementia care. Researchers will visit case study hospitals over 3 days, guided by the Edbrooke-Childs *et al* Huddle Observation Tool for inpatient clinical wards (please see below for more details on observations of staff on ward).

### Recruitment and data collection in case study sites

Case study site visits will take place over a 3-day period to undertake interviews, focus groups and observational data collection (for more detail see figure 3 and table 1).

### Observations of staff on ward

We will observe staff activity on wards at up to two staff shift changeovers, focusing on their administrative/general duties and the work environment. For example, before the start of the shift, or observation of domestic staff and porters at meals times. The observations will be guided by the Edbrooke-Childs *et al*'s[27] Huddle Observation Tool for inpatient clinical wards. This approach explores four main areas on the ward in relation to staff general/administrative duties:

1. 'Risk management' (eg, 'Were there opportunities to identify risks and come up with concrete plans for these risks?').
2. 'Structure' (eg, 'Did the activities have a clear structure?').

**Table 1** Case study site selection considerations and sample matrix

| Hospital | NAD score* | CQC scores† | Score discrepancy (NAD vs CQC) | Patient experience Friends and Family Test score: inpatient | Friends and Family Test score: outpatients | Staff experience Friends and Family Test score: staff—work | Dementia specialist ward in hospital | Dementia lead present at either ward or directorate level | Geographical region (ie, sustainability and transformation plans region) |
|---|---|---|---|---|---|---|---|---|---|
| Site 1 | Including a range of scores: ▲ High ▲ Mixed ▲ Low | Including a range of scores: ▲ Outstanding ▲ Good ▲ Requires improvement ▲ Inadequate | We aim to recruit a range of hospitals where there is 'score discrepancy' and where there is 'no score discrepancy' between the NAD and CQC scores. Example 1 (discrepancy): NAD score is high, yet CQC score is inadequate. Example 2 (no discrepancy): NAD score is low, and CQC score is inadequate. | Mix of high and low scores: ranging between 50% and 100% | | | Yes or no | Yes or no | For example: ▲ North ▲ Midlands and East ▲ South West |
| Site 2 | | | | | | | | | |
| Site 3 | | | | | | | | | |
| Site 4 | | | | | | | | | |
| Site 5 | | | | | | | | | |
| Site 6 | | | | | | | | | |

*NAD: National Audit of Dementia: the scores are based on 24 hospitals randomly sampled from stratified lists of hospitals scoring low, mixed and high across three main domains of dementia care (governance, staff training and patient care) derived from the NAD in 2016 for our staff survey sample selection.
†CQC: Care Quality Commission: the score will be based on the Hospitals overall CQC score report: https://www.cqc.org.uk/

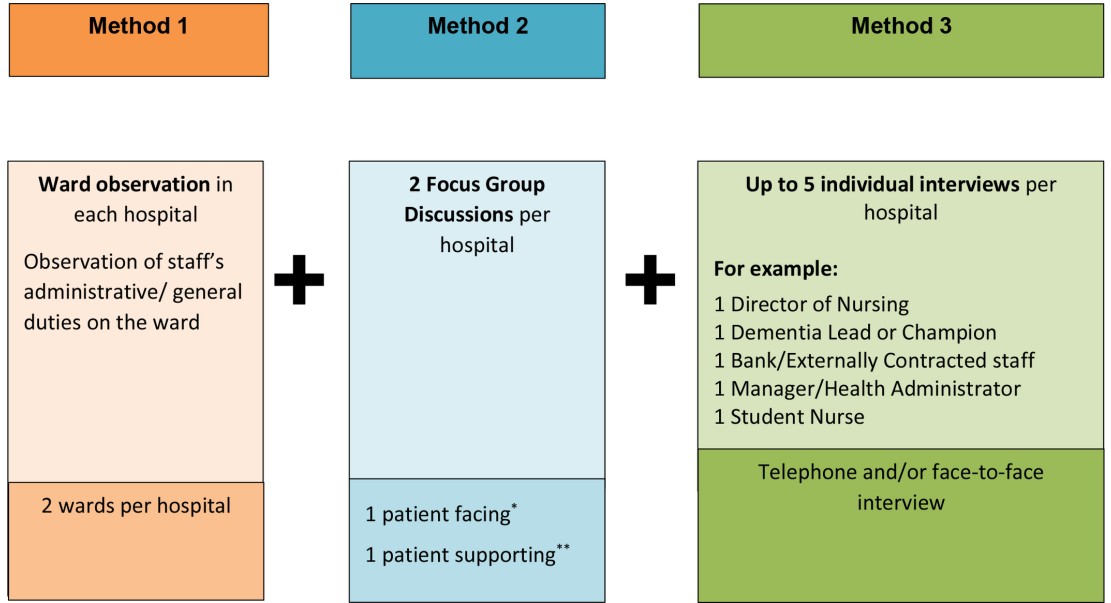

**Figure 3** Proposed data collection structure in each of the six hospitals. *Patient facing: doctors, nurses, healthcare assistants, allied health professionals, including outpatient, A&E, X-rays, dietitians, frailty ward staff, student nurses. **Patient supporting: social care, voluntary (community and hospital), discharge staff, patient advice and liaison service staff, car park, and security staff, porters, and other domestic and support staffs.

3. 'Collaborative culture' (eg, 'Did everyone have the opportunity to contribute and were all points of view respected?').
4. 'Environment' (eg, Physical and emotional environment).

The Huddle Observation Tool was developed as an observational assessment tool to assess the team processes occurring during huddles, including the effectiveness of hospital huddles to provide structured case management discussions to improve situation awareness on inpatient clinical wards.[27] It, therefore, provides a systematic manner to explore the interaction between staff within a hospital ward environment. Moreover, any relevant observations that do not fit the four areas (risk management, structure, collaborative culture and environment) of the Huddle Observation Tool, can be added as additional notes. Researchers will only make observation notes during a visit. Observations will not be connected to a specific individual or staff member. It will be made clear and staff will be reassured, that the aim of observation of administrative and general duties is to identify best practice rather than highlight faults. Permissions to conduct the observation on a ward will be sought from the location navigator (eg, ward managers/dementia leads), except for in a public space (eg, main entrance or reception area). All researchers will have appropriate approvals from local Research and Development (R&D) and the Health Research Authority (HRA), including relevant training (eg, research passport and good clinical practice).

*Interviews and focus groups of hospital staff*

We will aim to conduct two focus groups per hospital (total: 12 groups) with a range of staff—group 1 will be with 'patient facing staff' and group 2 will be with 'patient supporting' staff. Location navigators and CRN will help to identify staff and distribute information sheets and consent forms. Participants will be selected purposively to allow a maximum of 10 participants per focus group:

1. *Patient facing (group 1)*: Doctors, nurses, healthcare assistants, allied health professionals, including outpatient, Accident and Emergency (A&E), X-rays, dietitians, frailty ward staff, student nurses.
2. *Patient supporting (group 2)*: Social care, voluntary (community and hospital), discharge staff, patient advice and liaison service staff, car park, and security staff, porters and other domestic and support staff.

Our preliminary discussions with our advisory group (including local dementia leads responsible for ensuring that their hospital is aligned with NHS Trust and National dementia strategies, and involved in developing and implementing dementia training in their hospitals) indicated that separating 'patient facing' staff from 'patient supporting' staff would facilitate greater discussion of dementia training. Basic dementia knowledge (due to clinical training) and access to training is likely to differ considerably between the two groups, which may mean that some non-clinical staff (eg, domestic staff) may be hesitant to talk about their dementia knowledge and access to dementia training in front clinical staff.

We will conduct up to five interviews per hospital with a range of staff, including:
► Director of nursing.
► Dementia lead or champion.
► Bank/externally contracted staff.
► Manager/health administrator.
► Student nurse.

We estimate each interview and focus group to last on average an hour and will be audio recorded on encrypted

devices. The group will be shown a summary of their hospital's performance in seven areas, which is currently available from NAD. This will encourage greater engagement and interaction between the participants and serve to prompt discussion. We expect the focus groups to take place at the hospitals.

## Analysis of data

### Quantitative analysis

We will use Stata to analyse the quantitative data collected. The main focus will be to examine how hospital training scores relate to the difference in outcomes between patients with a known diagnosis of dementia, compared with outcomes for patients without a known diagnosis. We will develop a number of statistical models to determine the types and elements of training that are most strongly associated with changes in the LoS, ERs within 30 days of prior discharge, outcomes related to the care received in the hospital for people living with dementia and mortality (death within 30 days of discharge) for older people. We will use NAD data (2010/2011, 2012/2013 and 2016/2017) to construct hospital-level scores on a number of key domains of dementia care, in particular governance, staff training and patient care. Statistical models will be fitted to assess relationships between scores on the different domains of dementia care and differences in outcomes between people living with dementia compared with those without, after controlling for a range of confounding variables including patient demographics, pre-existing health and reasons for hospital admission. A further analysis will examine relationships between outcomes and dementia care scores derived from our own hospital organisational survey in 2016/2017 only.

### Qualitative analysis

A thematic analysis[28] will be undertaken to identify emerging themes (including a priori themes, in view of the literature and current dementia policies) in relation to case study interviews, focus groups and observations. Data will be imported, coded and managed using the qualitative data analysis software tool (NVivo V.11). The analysis process will be guided by our programme theory that is being developed as part of a systematic review (ie, critical interpretive synthesis) of the current evidence base on dementia training in hospital settings.[18]

### Economic analysis

A combined decision analytical/simulation model will be developed from the phases 1 and 3 data and review of the literature. The economic evaluation will use data from phases 1 to 3 to develop the model structure and explore the costs, outcomes and relative cost-effectiveness of alternative training interventions from the perspective of health and social care services and patients. The primary measure of benefit will be the quality-adjusted life year measure. Incremental cost-effectiveness ratios will be calculated. Cost-effectiveness acceptability curves, the probability of whether training is cost-effective and net benefit statistics

will be estimated to explore uncertainty in the data. One and multiway sensitivity analyses will be used to explore the impact of alternative training methods over alternative time horizons (eg, 5 and 10 years), patient populations and settings, alternative data sources and range of costs are included. This will include assessment of the impact of barriers and facilitators to implementing training and staff participation and compliance with training.

## DEMTRAIN PATIENT AND PUBLIC INVOLVEMENT AND CO-RESEARCH MODEL

There is increasing recognition that inclusion of people living with dementia beyond the realm of 'participant' to involvement in all areas of the research process in ways which are both personally meaningful and relationally supportive is imperative.[29] The CO-research INvolvement and Engagement in Dementia (COINED) model[30] forms part of the work of the Neighbourhoods and Dementia Study and is a unique and positive feature of the DEMTRAIN study. The term 'co-researcher' reflects collaborative, cooperative and community-based partnership between groups of people living with dementia, academic researchers and service providers. We will involve people living with dementia and their care partners in a range of areas including:

► Development of the organisation and staff level survey.
► Development of qualitative interview and focus group guide.
► Regular feedback on study design and implementation.

Care partners' is the term selected by co-research involvement groups of our Neighbourhoods Study, to refer to those who look after, support and care for someone living with dementia, in a non-professional, non-paid capacity.[31] This may be a family member, friend or neighbour.

The qualitative analysis of case study data provides an additional opportunity to bring in the perspectives of people living with dementia and their care partners at the analysis stage as co-researchers. We, therefore, propose various activities that will help in the interpretation of the data/findings and help us bring their valuable perspectives:

► Co-researchers will inform the development of the DEMTRAIN programme theory, that is, part of a systematic review (ie, critical interpretive synthesis) of the current evidence base on dementia training in hospital settings, and that serves as theoretical framework to our study.
► Co-research observational visits to hospital public spaces (eg, main reception and public eating areas), in up to six sites, with researchers who are involved in qualitative data collection for case studies.
► Co-researchers will review various anonymised iterations of the analysed summary/data.

All co-researchers will receive previsit and postvisit briefing before each hospital visit to discuss any issues or challenges concerning their observations. They will be accompanied by a researcher at all times and, if they wish, an informal care partner may escort them during the visit.

Given the nature of the research and data collection, we envisage no risk to researchers. However, WP8 provides a well-being service for all programme staff, co-researchers and members working within the Neighbourhoods Study. WP8 is led by RE, clinical psychologist, who provides this confidential support service.

## ETHICS AND DISSEMINATION

We plan to develop both standard (eg, academic publications, presentations at conferences) and innovative (eg, citizen scientist web portals, online fora, links with hospitals and third sector organisations) means of ensuring the study findings are accessible and disseminated regionally, nationally and internationally.

### Ethical considerations

Given the nature of the topics and methods adopted, the study poses a low risk. However, we understand that hospital staff may be anxious or fearful of reprisal from other colleagues or hospital management when completing a survey, such as our hospital staff survey, that explores their skills, competencies and satisfaction in regard to dementia training (particularly if the training is developed and provided by their own hospital). To manage these risks, all participant answers will be separate from any identifiable information, and we will adhere to Lancaster University guidelines on confidentiality and anonymity of participants. A key issue concerning staff observation is whether the individual concerned is 'identifiable' from the information collected (ie, in the observation notes). However, we will exclude any identifiable information in the notes and focusing only on administrative/general duties and the work environment.

Information provided to staff invited to complete a survey will ensure they are aware of this anonymity and that they are under no obligation to take part in the study. Participants will also be made aware that they are free to withdraw from a survey/interview/focus group if they wish. Staff will be informed that once a survey/interview/focus group is complete they have 2 weeks to withdraw the survey by submitting a request to the research team. Consent to participate in the staff survey will be taken as granted on completion of the survey. Before each interview and focus group, a researcher will go through the study information sheet and seek informed consent for each participant. Staff will be made aware that they are free to leave an interview or focus group if they wish. The consent forms will ensure that staff are aware that their participation is confidential, and that we will be audio recording the discussions.

Any quotes or reporting will not identify individual staff or their hospital. Audio files will be stored safely and securely in line with university and our study data management plan. For instance, all digital data (ie, audio recordings) will be stored on encrypted audio recorders (and backed up at the earliest opportunity onto encrypted hard drives and subsequently onto secure university servers. Only the research team and those involved in transcribing the data will have access to the raw data. All participant information will be stored in locked cabinets.

All staff participating in the study (ie, staff survey, interview and focus group participants) will be offered a £10 voucher as acknowledgement of their contribution and time. Similarly, continuing professional development (CPD) certificates for participation in research will be provided to all staff members who indicate that they wish to receive one. As thanks for participation in the study, we will offer a Dementia Clock to the hospital wards taking part in the staff survey (up to two Dementia Clocks per hospital). Summary reports based on the primary data collection and secondary data (ie, HES and NAD data) will be made available to all hospitals who complete the hospital organisational survey and indicate that they wish to receive one. Given individual staff responses will be confidential this summary will not include information relating to the staff survey. Completed organisational surveys will be entered into a prize draw for up to 10 hospitals to receive £500 for either equipment that enhances the hospital experience for patients living with dementia or a dementia-related charitable donation.

### Strengths and limitations

This study will use the principles of programme theory to expand on mechanisms and the interactions between them, facilitators and barriers to dementia training in acute hospitals in England. One of the key strengths of this study is that it uses a range of existing datasets (including HES and NAD and mixed-methods approaches to make recommendations in relation to dementia training in acute hospital settings. Unlike previous studies that have focused on immediate, on-the-day training feedback and effects of dementia training in hospital settings,[10 11] our proposed study will attempt to explore the longer-term impact of dementia training. Given the multiple initiatives and dementia policies employed in hospitals across England, it will be challenging to establish the effects of dementia training directly on patient and staff outcomes. Nonetheless, we will use statistical models and mixed-methods approaches to adjust and triangulate our findings.

Another key strength of our study is that across each of the study phases, we will continue to facilitate the involvement of people living with dementia as co-researchers, guided by the COINED model of co-research. As well as, consultations with our collaborators and key stakeholders involved in similar form of studies, such as What Works' in Dementia Education and Training evaluation (http://www.leedsbeckett.ac.uk/pages/what-works/), NAD and local CRN. Our project partner, Advancing Quality Alliance an improvement body working across the NHS in the North West will also be consulted to aide recruitment and dissemination of the findings.

**Acknowledgements** This work forms part of the Economic and Social Research Council (ESRC) and the National Institute for Health Research (NIHR)

Neighbourhoods and Dementia mixed-methods study (https://sites.manchester.ac.uk/neighbourhoods-and-dementia/) under grant: ES/L001772/1. This paper is taken from Work Programme 5, Developing the evidence base for evaluating dementia training in NHS hospitals (DEMTRAIN). Our special thanks to Garstang Memory Café, Scottish Dementia Working Group, all of our co-researchers who have helped us, and continue to be an integral part of the DEMTRAIN study. We thank the participants and all who helped us in the study, especially the DEMTRAIN Advisory Group, Carol Opdebeeck, Manchester Metropolitan University and Marie Crane, Research Coordinator.

**Contributors** SR, DR, LD and JK designed the DEMTRAIN study outline submitted to NIHR/ESRC. FA is the lead researcher on the study and led on developing the protocol for this publication. AH and HM are current researchers on the study. SR is the principal investigator for this work programme. JK is the chief investigator for all eight of the Neighbourhoods and Dementia study work programmes. DR, CS, LD, MH, FH, RE, IL, SB and AB are part of the wider team and provide wider expertise. All authors contributed to the development of the study protocol and this publication.

**Funding** This study was funded jointly by the Economic and Social Research Council (ESRC) and the National Institute for Health Research (NIHR) under grant: ES/L001772/1. The ESRC is part of UK Research and Innovation.

**Disclaimer** The views expressed are those of the author(s) and not necessarily those of the ESRC, UKRI, NHS, the NIHR or the Department of Health and Social Care.

**Competing interests** None declared.

**Patient consent for publication** Not required.

**Ethics approval** The study has been reviewed and approved by the Faculty of Health and Medicine Research Ethics Committee (FHMREC 17056) and HRA (IRAS ID 242166: REC reference 18/HRA/1198).

**Provenance and peer review** Not commissioned; externally peer reviewed.

**ORCID iDs**
Faraz Ahmed http://orcid.org/0000-0001-6714-1500
Linda Davies http://orcid.org/0000-0001-8801-3559
Siobhan Reilly https://orcid.org/0000-0003-4372-4415

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
