## [Reviewer comments · BMJ Open]

ARTICLE DETAILS

TITLE (PROVISIONAL)	Developing the evidence base for evaluating dementia training in NHS hospitals (DEMTRAIN): a mixed methods study protocol
AUTHORS	Ahmed, Faraz; Morbey, Hazel; Harding, Andrew; Reeves, David; Swarbrick, Caroline; Davies, Linda; Hann, Mark; Holland, Fiona; Elvish, Ruth; Leroi, Iracema; Burrow, Simon; Burns, Alistair; Keady, John; Reilly, Siobhan

VERSION 1 - REVIEW

REVIEWER	Dr Melanie Handley University of Hertfordshire, UK
REVIEW RETURNED	07-Jun-2019

GENERAL COMMENTS	Thank you for the opportunity to review the DEMTRAIN study protocol. This is a well written manuscript and the study findings should lead to useful recommendations for dementia training in hospitals. Below are suggested revisions for the authors. Page 5/6: The tense for reporting the methods is not consistent, changing from future to past tense. "Recruitment for organisational survey in acute NHS hospitals in England" is in future tense and "Recruitment for staff survey in two specific wards in acute NHS hospitals in England" is in past tense. Please amend for consistency. Page 6 Line 33: Consider adding a line to clarify that where assent is not given, survey data will be deleted and not analysed. I realise this information is included in the figure, but it would be useful to reiterate this in the text. Page 8, Lines 40 to 60: The authors state they will use a realist approach (page 8, line 52/53). The subsequent reference to facilitators and barriers is not in line with the approach. In the Pawson and Tilley realist evaluation model, the focus is not to identify facilitators and barriers but instead to explain what has caused an outcome (i.e. the mechanism) and in what context the mechanism was triggered. Please amend as necessary. Page 9: Please provide a few sentences to clarify the rationale for the observations and the selection of the Huddle Observation Tool. Page 10, Lines 12 to 17: I read this paragraph several times as I was unclear whether the focus groups would combine staff from
--

	both staff groups or not. Please consider rewording and provide a rationale for mixed staff focus groups. Page 11: I was pleased to see the breadth of involvement of co-researchers with dementia on the study. Page 13 Line 56: There is an open bracket, please close or remove. I wish the authors all the best with their submission and the DEMTRAIN study. I look forward to reading the findings.
--	---

REVIEWER	Machiko Inoue Hamamatsu University School of Medicine, Japan
REVIEW RETURNED	28-Jun-2019

GENERAL COMMENTS	Thank you for the opportunity to review the manuscript. This is a protocol of very large-scale study, which comprehensively deals with dementia training for hospital staff. It has two main research questions and three phases, and uses multiple methods for each phase. I would recommend authors to describe more about: 1) how the results of each phase will be interpreted to answer the main two questions. - A diagram may be helpful to show the overview of the whole study and the relationship among three phases and different data sources. 2) the integration process of the three phases, and qualitative and quantitative results.
--

REVIEWER	Gillian Stockwell-Smith Adjunct Research Fellow Menzies Health Institute Queensland Griffith University Australia
REVIEW RETURNED	01-Jul-2019

GENERAL COMMENTS	Thank you for the opportunity to review your protocol, the study will address a topic of great interest and is an important step in recognising and responding to a pressing need to improve the care of people with dementia in acute hospital settings. I found the protocol coherent and well-structured and have only a one suggestion to make. The introduction would benefit from a sentence or two introducing/describing current concerns regarding the quality of care for people with dementia in acute care to justify the study focus on staff dementia training.
---

VERSION 1 – AUTHOR RESPONSE

Reviewer 1

1. Page 5/6: The tense for reporting the methods is not consistent, changing from future to past tense. "Recruitment for organisational survey in acute NHS hospitals in England" is in future tense

and “Recruitment for staff survey in two specific wards in acute NHS hospitals in England” is in past tense. Please amend for consistency.

Thank you for the comment, we have amended the manuscript to keep the text in future tense [Page 6, line 197-201 & Page 7, line 202-204].

2. Page 6 Line 33: Consider adding a line to clarify that where assent is not given, survey data will be deleted and not analysed. I realise this information is included in the figure, but it would be useful to reiterate this in the text.

Thank you for the comment, we have amended the manuscript [Page 6, line 194-195].

3. Page 8, Lines 40 to 60: The authors state they will use a realist approach (page 8, line 52/53). The subsequent reference to facilitators and barriers is not in line with the approach. In the Pawson and Tilley realist evaluation model, the focus is not to identify facilitators and barriers but instead to explain what has caused an outcome (i.e. the mechanism) and in what context the mechanism was triggered. Please amend as necessary.

We thank the reviewer for making the comment. We would like state that we are using principles of programme theory and realist approach (i.e. not realist evaluation); i.e. “We will utilise the principles of programme theory and realist approach to identify theories of change with respect to training initiatives in different hospital contexts 23 24. For instance to identify mechanisms and the interactions between them, as well as facilitators and barriers to dementia training in the hospital and its related outcomes.”

The discussion on programme theory has moved considerably since Pawson and Tilley realist approach. However, given the placement of references, we can see that it can be interpreted that Pawson and Tilley have suggested the emphasis on barriers and facilitators, we have therefore amended the text and moved references [Page 9, line 278-286]:

We will utilise the principles of programme theory and realist approach to identify theories of change with respect to training initiatives in different hospital contexts. For instance to identify mechanisms and the interactions between them to explain what has caused an outcome and in what context 24 25, as well as facilitators and barriers to dementia training in the hospital and its related outcomes 24. Programme theory, within the theory-driven evaluation field, refers to a variety of ways of describing and evaluating a programme or intervention (such as staff dementia training) by developing an understanding of the causal modal that connect the inputs and activities of an intervention to its outcomes 16.

4. Page 9: Please provide a few sentences to clarify the rationale for the observations and the selection of the Huddle Observation Tool.

We thank reviewer for raising the point and have now amended the manuscript in view of the comments [Page 11, line 326-332]:

The Huddle Observation Tool was developed as an observational assessment tool to assess the team processes occurring during huddles, including the effectiveness of hospital huddles to provide structured case management discussions to improve situation awareness on inpatient clinical wards 27. It, therefore, provides a systematic manner to explore the interaction between staff within a hospital ward environment. Moreover, any relevant observations that do not fit the four areas (risk management, structure, collaborative culture and environment) of the Huddle Observation Tool, can be added as additional notes.

5. Page 10, Lines 12 to 17: I read this paragraph several times as I was unclear whether the focus groups would combine staff from both staff groups or not. Please consider rewording and provide a rationale for mixed staff focus groups.

We thank the reviewer for highlight this, and have amended the manuscript to clarify that we are not mixing the staff, and added the rationale for our choice [Page 11, line 344-347 and Page 12, line 348-363]:

We will aim to conduct two focus groups per hospital (total: 12 groups) with a range of staff – Group one will be with ‘patient facing staff’ and Group two will be with ‘patient supporting’ staff. Location navigators and CRN will help identify staff and distribute information sheets and consent forms. Participants will be selected purposively to allow a maximum of 10 participants per focus group:

1. Patient facing (Group 1): Doctors, nurses, health care assistants, allied health professionals, including outpatient, A&E, X-rays, dietitians, frailty ward staff, student nurses.
2. Patient supporting (Group 2): Social Care, Voluntary (Community and Hospital), Discharge Staff, Patient Advice and Liaison Service staff, Car Park, & Security Staff, Porters, and other Domestic and support staff

Our preliminary discussions with our advisory group (including local Dementia Leads responsible for ensuring that their hospital is aligned with NHS Trust and National dementia strategies, and involved in developing and implemented dementia training in their hospitals), indicated that separating ‘patient facing’ staff from ‘patient supporting’ staff would facilitate greater discussion of dementia training. Basic dementia knowledge (due to clinical training) and access to training is likely to differ considerably between the two groups, which may mean that some non-clinical staff (e.g. domestic staff) may be hesitant to talk about their dementia knowledge and access to dementia training in front clinical staff.

6. Page 11: I was pleased to see the breadth of involvement of co-researchers with dementia on the study.

We thank the reviewer for acknowledging the breadth of involvement of our co-researchers in our study.

Reviewer 2

I would recommend authors to describe more about:

1. how the results of each phase will be interpreted to answer the main two questions.
- A diagram may be helpful to show the overview of the whole study and the relationship among three phases and different data sources.

We thank the reviewer for raising this point and can confirm that for each of the three phases we have specific aims included (on page 5,7 and 8), which address our study's two overarching research questions (on page 4). In addition, we have now also included figure 1 that shows the various level at which integration process occurs.

2. the integration process of the three phases, and qualitative and quantitative results.

We have expanded more on the integration process using figure 1, and amended page 5 line 138-139.

Reviewer 3

1. I found the protocol coherent and well-structured and have only a one suggestion to make. The introduction would benefit from a sentence or two introducing/describing current concerns regarding the quality of care for people with dementia in acute care to justify the study focus on staff dementia training.

We thank the reviewer for their comments have included further references to the introduction to discuss concerns regarding the quality of care for people with dementia in acute care to, in addition to what has been already been included [Page 4, line 91-106]:

Improving the quality of care in general hospitals continues to be one of the key priorities within national dementia strategies 1-3. Acute NHS Trusts provide a range of services, including accident and emergency departments, inpatient and outpatient medicine and surgery and in some cases very specialist medical care 4. They provide secondary care, ranging from relatively small district hospitals to large city teaching hospitals in England. The term acute, generally refers to physical illnesses and conditions, which are usually short-term and require diagnostic tests, treatment and follow-up care 4 5. At any one time, around 70 percent of acute hospital beds are occupied by older people, 40% of whom have dementia 6-8. Despite limited literature in the area, a recent ethnography of the care

received by people living with dementia in acute hospital suggest that staff struggle to respond to the needs of people living with dementia in acute care settings and that training may be one of the intervention that can help to address quality of care issues observed 9. Dementia awareness and care training is widely thought to increase staff competencies 10 11, although types of training and how effectiveness is assessed is highly variable.

VERSION 2 – REVIEW

REVIEWER	Dr Melanie Handley University of Hertfordshire, UK
REVIEW RETURNED	10-Oct-2019

GENERAL COMMENTS	I would like to thank the authors for making the revisions. This is an important study and I look forward to publication of the findings.
---